# Feasibility of Adjusting the S_2_O_3_^2−^/NO_3_^−^ Ratio to Adapt to Dynamic Influents in Coupled Anammox and Denitrification Systems

**DOI:** 10.3390/ijerph17072200

**Published:** 2020-03-25

**Authors:** Yuqian Hou, Shaoju Cheng, Mengliang Wang, Chenyong Zhang, Bo Liu

**Affiliations:** State Key Laboratory of Pollution Control and Resource Reuse Research, School of the Environment, Nanjing University, Nanjing 210046, China; hyq_miao@163.com (Y.H.); chengshaoju@yeah.net (S.C.); Maywml1995@163.com (M.W.); zcylodge@163.com (C.Z.)

**Keywords:** Anammox, coupled denitrification, S/N ratio regulation strategy, total nitrogen removal, competition and cooperation

## Abstract

In this study, anammox, sulfur-based autotrophic denitrification, and heterotrophic denitrification (A/SAD/HD) were coupled in an expanded granular sludge bed (EGSB) reactor to explore the feasibility of enhancing denitrification performance by adjusting the S_2_O_3_^2−/^NO_3_^−^ (S/N) ratio to accommodate dynamic influents. The results indicated that the optimal influent conditions occurred when the conversion efficiency of ammonium (CEA) was 55%, the S/N ratio was 1.24, and the chemical oxygen demand (COD) was 50 mg/L, which resulted in a total nitrogen removal efficiency (NRE) of 95.0% ± 0.5%. The S/N ratio regulation strategy was feasible when the influent COD concentration was less than 100 mg/L and the CEA was between 57% and 63%. Characterization by 16S rRNA sequencing showed that *Candidatus Jettenia* might have contributed the most to anammox, while *Thiobacillus* and *Denitratisoma* were the dominant taxa related to denitrification. The findings of this study provide insights into the effects of CEA and COD on the performance of the A/SAD/HD system and the feasibility of the S/N ratio regulation strategy.

## 1. Introduction

Nitrogen contamination is a phenomenon that occurs in the global water environment system. Many new technologies for the removal of total nitrogen (TN) have been recently investigated, such as partial nitrification; Single reactor systems for High Ammonium Removal Over Nitrite (SHARON); anammox; and aerobic/anoxic deammonification, among which anaerobic ammonium oxidation (anammox) is one of the most promising approaches [1]. Without the addition of organic carbon, anammox can convert ammonium (NH_4_^+^) to nitrogen (N_2_) by using nitrite (NO_2_^−^) as an electron acceptor under anoxic conditions; compared to conventional nitrogen removal processes of nitrification/denitrification, anammox prominently reduces aeration and completely exempts the demand of exogenous organic carbon source, which saves costs for resources and energy consumption [2,3,4]. The reaction is as follows [5]:NH_4_^+^ + 1.32NO_2_^−^ + 0.066HCO_3_^−^ + 0.13H^+^ → 1.02N_2_ + 0.26NO_3_^−^ + 0.066CH_2_O_0.5_N_0.15_ + 2.03H_2_O(1)

However, nitrite is unstable in the environment and exists at a relatively low concentration in wastewater, limiting the anammox reaction [6,7]. It has been frequently reported that sulfur-based autotrophic denitrification (SAD) can provide nitrite for anammox through the conversion of nitrate into nitrite [8]. Thiosulfate has been proven to be the most readily utilized electron donor with the highest nitrate degradation rate in several studies [9,10]. Therefore, it has been treated as a substitute for sulfur pollutants in experiments conducted for the purpose of fast enrichment of SAD bacteria. As shown in the reactions in Equations (2) and (3), since Equation (2) occurs in preference to Equation (3) [11], the S_2_O_3_^2−/^NO_3_^−^ (S/N) ratio is a crucial factor for nitrite accumulation in the A/SAD system. In one recent study coupling autotrophic denitrification with partial nitritation-anammox (PNA) for efficient total inorganic nitrogen removal, the integrated treatment scheme removed up to 97% of the total incoming inorganic nitrogen [12]. Another study showed that, when the influent contained sulfide, approximately 70% of nitrate was transformed into nitrite in a reactor with an S/N ratio of 1:0.76 and the sulfide removal rate reached 90% [13]. These studies demonstrated that the A/SAD system can achieve a significant TN removal efficiency by adjusting the S/N ratio.

However, in actual sewage containing organic matter, heterotrophic denitratation (NO_3_^−^→NO_2_^−^) can also produce nitrite, which, together with ammonium, can be converted to dinitrogen gas by anammox [14]. However, heterotrophic denitrifiers are more energetically active (−427.0 kJ/mol) with higher growth rates (0.25/h) and biomass yields (1.1 gVSS/g NO_3_^−^-N) than anammox bacteria (−355.0 kJ/mol, 0.0075–0.014/h and 0.1 gVSS/g N, respectively) [14]. It is important to determine the impact of organic matter on the A/SAD system. Sodium acetate is a common carbon source for denitrification (Equation (4)).
S_2_O_3_^2−^ + 2.626NO_3_^−^ + 0.043CO_2_ + 0.644HCO_3_^−^ + 0.137NH_4_^+^ + 0.631H_2_O → 0.137C_5_H_7_O_2_N + 2.626NO_2_^−^ + 1.494H^+^ + 2SO_4_^2−^(2)
S_2_O_3_^2−^ + 2.070NO_2_^−^ + 0.028CO_2_ + 0.419HCO_3_^−^ + 0.089NH_4_^+^ + 0.400H^+^ → 0.089C_5_H_7_O_2_N + 1.035N_2_ + 0.275H_2_O + 2SO_4_^2−^(3)
CH_3_COO^−^ + 8/5NO_3_^−^ → 4/5N_2_ + 2CO_2_ + 1/5H_2_O + 13/5OH^−^(4)

Theoretically, anammox and denitrification coexist symbiotically in an actual engineering context and they occur in similar ecological environments. Anammox produces approximately 11% NO_3_^−^-N [15], and denitrification can remove NO_3_^−^ from anammox and supply anammox bacteria with nitrite, which achieves the goal of removing TN and chemical oxygen demand (COD) (Figure 1). However, the COD and conversion efficiency of ammonium (CEA) of sewage fluctuate, and it is significant to analyze the appropriate S/N ratio for maintaining the maximum nitrogen removal efficiency (NRE).

Knowledge on the cooperative mechanisms between anammox and denitrification in different influent conditions is still unclear, and more research is needed to be done. This study established an A/SAD/HD system to achieve the optimal NRE of the reactor. The influent of CEA and COD on the performance of A/SAD/HD system and the feasibility with an S/N regulation strategy based on speculation of cooperation between anammox and denitrification were investigated. Additionally, high-throughput sequencing of 16S rRNA gene amplicons was conducted to profile the change in the bacterial community and to identify the structuring of microbial community, and results were made relevant to reactor operation conditions. 

## 2. Materials and Methods

### 2.1. Operation of the Expanded Granular Sludge Bed Reactor (EGSB) Reactor

In this study, a lab-scale EGSB reactor (approximate volume of 2 L, with a height of 400 mm and an inner diameter of 80 mm) was used to establish the A/SAD/HD system (Figure 2). The EGSB reactor was equipped with a water bath temperature control system and was continuously operated at 32 ± 2 °C; the hydraulic retention time (HRT) was maintained at 3 h. Synthetic wastewater containing ammonium sulfate, sodium nitrate, sodium acetate, sodium thiosulfate, and other macro and microelements [4] was fed into the reactor. Components and concentration of macro and microelements (mg L^−1^): KH_2_PO_4_ 27.2, CaCl_2_·2H_2_O 180, MgSO_4_·7H_2_O 300, KHCO_3_ 500; and 1.25 mL L^−1^ trace element solutions A and B. Trace element solution A contained (g L^−1^): Ethylene Diamine Tetraacetic Acid (EDTA) 5 and FeSO_4_ 5; Trace element solution B contained (g L^−1^): EDTA 15, H_3_BO_4_ 0.014; MnCl_2_·4H_2_O 0.99; CuSO_4_·5H_2_O 0.25; ZnSO_4_·7H_2_O 0.43; NiCl_2_·6H_2_O 0.19; NaSeO_4_·10H_2_O 0.21; NaMoO_4_·2H_2_O 0.22; and CoCl2·6H_2_O 0.24.

The initial volume ratio of anammox sludge and denitrifying sludge for mixing was 3:1. The mixed liquor volatile suspended solid (MLVSS) of the EGSB reactor after integration was about 2.15 g/L. The anammox sludge was obtained from a lab-scale anammox EGSB reactor, which was first set up to enrich the anammox bacteria with hydraulic retention time of 3 h and the temperature of 32 ± 0.2 °C; this process lasted for 208 days, and the mature denitrifying sludge enriched for 2 months in a lab-scale Upflow Anaerobic Sludge Blanket (UASB) reactor. 

After sludge inoculation (a week adaptive phase), the EGSB reactor was operated for 120 and 54 days at stage I and stage II, respectively. For the convenience of calculation, the TN concentration was 426 mg/L, and NO_3_^−^-N in the S/N ratio is the sum of the NO_3_^−^-N in the influent and the NO_3_^−^-N produced by anammox (preset influent NH_4_^+^-N was completely removed by anammox).

The stage I reaction was divided into four phases (Table 1). Stage I tested the operational effect of the A/SAD/HD system in the treatment of wastewater with various organic concentrations (0–150 mg/L). Additionally, the S/N ratio was regulated to achieve the maximum nitrogen removal efficiency and the minimum S/N ratio. The operating period of the A/SAD/HD process at stage I was divided into 4 phases (0–120 days) according to the different CODs and S/N ratios of the influent, with each phase lasting for 30 days. The S/N ratios in Table 1 were calculated by using the stoichiometric formula of SAD and heterotrophic denitrification as a function of the COD concentration.

To investigate the feasibility of increasing the nitrogen removal rate, the S/N ratio regulation strategy was applied under different CEA conditions (59%–65%) in the A/SAD/HD system. Hence, stage II was divided into nine phases (as shown in Table 1), including phases I, I′, II, II′, III, III′, IV, IV′, and IV″. Phases I, II, III, and IV are the control group of phases I′, II′, III′, and IV′, respectively. During this stage, the CEA increased from 59% to 65% with a COD concentration of 50 mg/L, and the S/N ratio was adjusted to respond to changes in the CEA. According to Equations (1)–(3), the CEA of the influent increased in a stepwise manner, followed by an increase in the influent S/N ratio with a ΔS/ΔN of 1.1:1 (molecular weight). Here, ΔN was defined as the sum of influent −1.32ΔNH_4_^+^-N and ΔNO_3_^−^-N.

### 2.2. Batch Experiments

Batch experiments were performed at the end of stage I and stage II to evaluate the underlying mechanisms of cooperation between anammox and denitrification in this A/SAD/HD system. Batch experiments were conducted in a serum bottle with a reaction volume of 250 mL under constant temperature conditions (32 ± 2 °C). The experimental parameters are divided into two groups; around 200 mL initial substrate with the same components and with a half concentration of the influent of phase II (stage I) and phase III′ (stage II) was added to the two groups, respectively. To ensure the effective removal of the substrate, the entire reaction lasted for 3 h. Additionally, the effluent was sampled at 0, 20, 40, 60, 80, 100, 120, 140, 160, and 180 min for chemical analyses.

### 2.3. Effluent Sampling and Chemical Analysis Methods

Effluent samples were collected from the reactor in triplicate every 72 h (stage I) or 24 h (stage II) and filtered with a glass fiber filter (0.22-μm pore size made in Sartorius, Gottingen, Germany) for chemical analysis.

Ammonium, nitrite, and nitrate were analyzed by colorimetric methods using spectrophotometer (UNIC, UV-2800, Danding International Trade Co., Ltd., Shanghai, China) [16]. Mixed liquor volatile suspended solids (MLVSSs) were determined using gravimetric method [17]. COD was determined using the dichromate-reflux colorimetric method [18]. S_2_O_3_^2−^ was analyzed by ion chromatography (ICS-1000) using a Dionex Ionpac column (AS19 column) in Sunnyvale, CA, USA.

The calculations performed were as follows:

(1) Nitrite accumulation efficiency (NAE):NAE = (1.32 (NH_4_^+^-N_inf_ − NH_4_^+^-N_eff_) + NO_2_^−^-N_eff_) / (NO_3_^−^-N_inf_ − NO_3_^−^ -N_eff_ + 0.26 (NH_4_^+^-N_inf_ − NH_4_^+^-N_eff_)) × 100%(5)

(2) Contribution rate of anammox to TN removal (CRA) and Contribution rate of denitrification to TN removal (CRD):CRA = (1 + 1.32 − 0.26) × (NH_4_^+^-N_inf_ − NH_4_^+^-N_eff_) / (TN_inf_ − TN_eff_) × 100%(6)
CRD = 1 − CRA(7)

In these expressions, A_inf_ and A_eff_ represent the substrate concentrations in the influent and effluent. The coefficients of 1, 1.32, and 0.26 in the equation represent the coefficients of the stoichiometric formula for anammox.

### 2.4. Sludge Sampling and DNA Extraction

The reactor sludge was sampled from the bottom of the reactor at the end of phase I (stage I), stage I, and stage II. The samples were immediately fixed with 100% ethanol at a ratio of 1:1 (*v*/*v*) and stored at −20 °C before DNA extraction. Total DNA was extracted using a Fast DNA Kit for Soil (MP Biomedicals, Santa Ana, CA, USA) following the manufacturer’s instructions. The extracted DNA was then subjected to quality inspection using a UV-Vis spectrophotometer (NanoDrop® ND-1000, Agilent, Santa Clara, CA, USA).

### 2.5. High-throughput Sequencing and Bioinformatic Analysis

In this experiment, the V4 hypervariable region of the bacterial 16S rRNA gene was selected for PCR amplification with the primer pair 515F (5’-GTGCCAGCMGCCGCGGTAA-3’) and 806R (5’-GGACTACHVGGGTWTCTAAT-3’). PCR amplification was performed according to the following program: denaturation at 95 °C for 2 min, followed by 25 cycles of denaturation at 95 °C for 30 s, annealing at 52 °C for 40 s, and extension at 72 °C for 90 s; a final extension was performed at 72 °C for 7 min. The amplicon was purified using a Wizard® SV gel and PCR product evolution system (Promega, Madison, WI, USA) and then sent to Shanghai Majorbio Biopharm Biotechnology Co., Ltd. (Shanghai, China) for high-throughput sequencing. The raw reads were pruned with FLASH and filtered with Trimmomatic software (version 0.39) (http://www.usadellab.org/cms/?- page = trimmomatic). Sequences that passed the quality check were classified into the operational classification units (OTU) at the 97% similarity level using the QIIME platform. Subsequently, the OTUs were classified at an 80% confidence threshold using the RDP (version 11.5) Classifier (http://rdp.cme.msu.edu/).

## 3. Results and Discussion

### 3.1. Operating Performance of the A/SAD/HD System in Stage I

Figure 3a shows the NRE of the A/SAD/HD reactor during the organic change stage. Without the addition of organic matter (phase I), the average NRE achieved in this reactor gradually reached 97.3% on the last five days of phase I, which is comparable to those of other A/SAD systems [14,19]. During phases II–IV, sodium acetate was fed to the reactor to promote heterotrophic denitrification. With an influent COD of 50 mg/L (phase II) or 100 mg/L (phase III), effluent NH_4_^+^-N, NO_2_^—^N, and NO_3_^−^-N accumulated at the beginning of each phase, resulting in a slight decrease in the NRE, possibly because HD bacteria require a certain adaptation period [20]. After adaptation, effluent NH_4_^+^-N, NO_2_^—^N, and NO_3_^−^-N decreased significantly in both phases II and III and the corresponding NRE reached 95.0% ± 0.5% and 93.6% ± 0.8% on the last five days of phases II and III, respectively. These results indicated that the S/N ratio regulation strategy could achieve optimal operation performance with influent COD concentrations of 50 mg/L and 100 mg/L in the A/SAD/HD system.

However, when excessive organic matter was added (influent COD of 150 mg/L during phase IV), effluent NH_4_^+^-N, NO_2_^—^N, and NO_3_^−^-N increased in the reactor and the average NRE was reduced to 80.1%. On the 118th day, the effluent NH_4_^+^-N concentration increased to 45.2 ± 2.1 mg/L, indicating the inhibition of anammox activity. There are two reasons for this effect. One is that the heterotrophic denitrifying bacteria exhibit a higher growth rate than anammox bacteria [21,22]; the microbial community analysis demonstrated that the relative abundance of *Candidatus Jettenia* decreased from 7.1% to 4.5% after sludge mixing (Figure 3b), and *Candidatus Jettenia* is an anammox bacteria [2], which affects the performance of its functions. The other is that COD replaces NH_4_^+^-N as an electron donor for anammox microorganisms [23,24]; as shown in Figure 3a, the increase in the NO_2_^−^-N concentration was less than the increase in NH_4_^+^-N, which was inconsistent with the stoichiometry of anammox.

Based on the above results, this study revealed that the A/SAD/HD system achieved a high NRE under low-influent-COD conditions (50 mg/L), that the S/N ratio regulation strategy was feasible when the influent COD concentration was less than 100 mg/L, and that the NRE of the reactor was slightly affected by the S/N ratio. Compared with the A/SAD system, partial denitrification was carried out by both heterotrophic and autotrophic denitrifying bacteria. COD is a pollutant in practical engineering, so this coupling system can meet the nitrogen removal and COD removal under the conditions of high ammonia nitrogen, high nitrate, and low C/N.

### 3.2. Operating Performance of the A/SAD/HD System in Stage II

In this study, the nitrogen-removal performance of the A/SAD/HD reactor was investigated with various influent CEA (stage II) and the S/N ratio regulation strategy was employed to optimize nitrogen removal. As shown in Figure 3b, increasing the CEA (from 59% to 65%) exerted a significant effect on the NRE in phases I, II, III, and IV, inducing the slight accumulation of nitrate and nitrite in the effluent. The accumulation of nitrite could inhibit anammox bacteria [25,26,27] and could further reduce nitrogen removal in this reactor. As a result, the effluent nitrite concentration rapidly reached 67.9 ± 1.4 mg/L on day 162, corresponding to the increase in ammonium in the effluent.

In terms of the accumulation of nitrate and nitrite in the effluent, the S/N ratio regulation strategy was applied to increase nitrogen removal in phases I′, II′, III′, and IV′. The increasing S/N ratio (from 1.24 to 1.3) exerted a significant effect on nitrogen removal in phase I′, inducing the effluent nitrite and nitrate concentrations to decrease to 8.3 ± 0.3 mg/L and 0 mg/L, respectively. Additionally, the NRE in phases I′, II′, and III′ reached greater than 96% with S/N ratio regulation. The S/N ratio normally changes the autotrophic denitrification process in the A/SAD system [28]; when the S/N ratio is low, thiosulfate is first used to reduce nitrate to nitrite. The increase in the S/N ratio was found to distinctly increase nitrogen removal in the reactor by promoting nitrite transformation through autotrophic denitrification. Additionally, ammonium in the effluent was reduced (from 24.5 ± 1.4 mg/L to 4.7 ± 0.7 mg/L in phase III′), indicating that the ammonium removal process was strengthened in the reactor. Previous studies have reported that anammox bacteria can recover activity after the removal of nitrite inhibition [29]. Similarly, the decrease in nitrite in the reactor exerted an inappreciable effect on anammox bacteria and was conducive to ammonium removal in the reactor.

It should be noted that the NRE was not improved by S/N regulation in phase IV′, and effluent ammonium and nitrite continuously accumulated in this study. When the S/N ratio was regulated to 1.42, average concentrations of 52.0 mg/L ammonium and 62.6 mg/L nitrite were achieved in the effluent. This result indicated that anammox was severely inhibited in this phase, which was principally caused by a high concentration of nitrite [30]. Additionally, the decreases in nitrate and nitrite demonstrated that more nitrate was directly transformed into nitrogen instead of nitrite due to the excess of thiosulfate. Therefore, as the CEA increased to 65% in the A/SAD/HD reactor, the S/N ratio regulation strategy was disabled to increase nitrogen removal. To eliminate the effect of CEA, the A/SAD/HD reactor was subsequently operated with a CEA of 55% (phase IV″), achieving stable nitrogen removal with an ammonium and nitrate removal efficiency of approximately 100%.

This study revealed that S/N ratio regulation optimized and increased nitrogen removal through autotrophic denitrification. However, this regulation strategy could be useful at a CEA lower than 65%. In other words, when the influent CEA reaches 65%, optimization of the cooperation of multiple nitrogen transformation processes (including autotrophic/heterotrophic denitrification and anammox) through regulating the S/N ratio is not an effective method. Therefore, further studies are necessary to explore new methods for increasing nitrogen removal in the A/SAD/HD system.

### 3.3. Cooperation and Competition between Anammox and Denitrification 

To evaluate the nitrogen metabolic pathways in the reactor, the contribution rates of anammox and denitrification to TN removal were characterized in different phases. As shown in Figure 4, the contribution rate of anammox to TN removal slightly increased (from 93.6% to 95.1%) with the addition of organic matter in stage I (phases I–III), which was consistent with other studies [31]. This indicated that autotrophic and heterotrophic denitrification processes mainly provided nitrite for anammox. Previous studies have demonstrated that partial denitrification can cooperate with anammox in an A/SAD system [12,13], and this cooperation relationship is further increased in an A/SAD/HD system. However, when the organic matter content was higher than 100 mg/L (phase IV in stage I), the contribution rate of anammox to TN removal markedly decreased to 91.9% and the decreasing NAE indicated that a decreasing amount of nitrite was transformed to nitrogen through denitrification. Previous report shows that high concentrations of COD can inhibit the activity of anammox bacteria in reactors [32] and that a high organic content facilitates the occurrence of complete denitrification. Although the organic content was not high in this study, a similar effect occurred: an intense competition between anammox and denitrification occurred in this A/SAD/HD system.

With the increase in the CEA in stage II (phases I, II, and III), the contribution rate of anammox also slightly fluctuated, decreasing from 88.5% to 85.7% (Figure 4). Due to the insufficient amount of ammonium in the reactor, the activity of anammox bacteria was restricted [33] and nitrite was transformed into nitrogen through the denitrification process. Consequently, thiosulfate and organic matter were utilized for nitrite reduction, inducing the accumulation of nitrate in the reactor (Figure 3b). In phases I′, II′, and III′, the increased S/N ratio increased denitrification. As expected, the contribution rate of denitrification to TN removal significantly increased. For instance, in phase II′, after the S/N ratio was increased from 1.24 to 1.34, the contribution rate of denitrification to TN removal increased from 11.7% to 15.3%. It should also be noted that more than 77% of TN was removed through anammox, indicating that anammox remained the primary nitrogen metabolic pathway in this A/SAD/HD system. However, as the CEA increased to 65% in phase IV (stage II), the contribution rate of anammox to TN removal obviously decreased to 27.8%. Additionally, S/N ratio regulation promoted the occurrence of complete denitrification in phase IV′ and the corresponding contribution rate of denitrification to TN removal increased to 74.7%. The serious insufficiency of ammonium in the reactor clearly limited anammox [33] to and facilitated nitrite reduction through denitrification. Additionally, previous studies have demonstrated that high nitrite accumulation inhibits anammox [27,34], which was also demonstrated by the observed nitrite accumulation in the effluent (Figure 3b). Therefore, the complete denitrification process was mainly responsible for nitrogen removal in this phase. As the CEA was reduced to 55% in phase IV″, anammox recovered, with a contribution rate of 89.9%.

To evaluate the temporal variation of the contribution rate to TN removal under different CEA conditions (55% and 63%), two batch experiments were conducted by culturing sludge for 300 min. As shown in Figure 5, the consumption of electron donors (COD and S_2_O_3_^2−^) was similar during the first 40 min. However, lower nitrite accumulation was detected under low CEA conditions (55%), indicating that nitrite was mainly converted to nitrogen through denitrification. As a result, the contribution rate of denitrification to TN removal reached 12.6% at 40 min (Figure 5c). After 100 min, nitrate produced by anammox was immediately reduced by denitrification. This further indicates a dynamic balance between anammox and denitrification. At this time, a high contribution rate of anammox to TN removal was achieved under low CEA conditions (Figure 5c), which is consistent with the nitrogen conversion characteristics in the A/SAD process reported by Reference [35]. When the CEA was 63%, after the organic matter was exhausted in the first 40 min (Figure 5c), the contribution rate of denitrification to TN removal decreased to 1.0% at 60 min (Figure 5d). This result reveals that heterotrophic denitrification reduces nitrate to nitrogen and that autotrophic denitrification preferentially converts nitrate to nitrite. It should be noted that the contribution rate of denitrification gradually increased to 52.0% at 160 min (Figure 5d), which also demonstrated the high activity of the complete denitrification process under high CEA conditions. Therefore, the complete denitrification process may be accomplished by autotrophic denitrifying bacteria. Based on the above results, the S/N regulation strategy may increase the TN removal efficiency in the A/SAD/HD system through autotrophic denitrification.

The results indicated that heterotrophic denitrification can increase the NRE when the organic concentration is not higher than 100 mg/L. Through cooperation between autotrophic denitrification and heterotrophic denitrification, the contribution rate of anammox to TN removal is improved. In low-ammonia nitrogen wastewater, the shortage of ammonium limits nitrite removal by anammox oxidation. The strengthening of denitrification by increasing the S/N ratio could improve the NRE.

### 3.4. Microbial Community Structure and Functional Bacteria Composition

The bacterial taxa showing relative abundances over 0.1% at the phylum and genus levels were characterized (Figure 6a,b).

The 16S rRNA gene sequencing results revealed that 11 predominant phyla were detectable in the reactor (Figure 6a), including Proteobacteria, Planctomycetes and Chloroflexi in particular. Specifically, the predominant phyla (Proteobacteria, Planctomycetes, and Chloroflexi) detected in this A/SAD reactor agree with previous studies [36,37]. After the addition of organic matter (A/SAD/HD system), the relative abundance of phylum Proteobacteria increased remarkably (from 38.1% to 49.0%); conversely, the relative abundance of Planctomycetes exhibited a decreasing trend (from 23.8% to 19.7%). Various heterotrophic denitrifying bacteria are affiliated with Proteobacteria [38], and the addition of organic matter actually promotes their enrichment in the autotrophic system [39]. In addition, anammox bacteria affiliated with Planctomycetes have been known to utilize inorganic carbon for growth, and organic matter significantly inhibits their enrichment [32]. In stage II, after S/N ratio regulation, the relative abundance of Proteobacteria continuously increased (from 49.0% to 61.7%), whereas the relative abundance of Planctomycetes showed a decreasing trend (from 19.7% to 14.6%) with the increase in the CEA. Additionally, Chloroflexi constituted the skeleton of anaerobic granular sludge [25], showing the same downward trend from 9.1% to 5.1%.

Figure 6b shows that the autotrophic denitrifying bacteria were dominated by *Thiobacillus* spp. (23.9%) in the A/SAD reactor, which provide nitrite for anammox bacteria in the A/SAD system [28]. Similarly, previous studies have also detected that *Thiobacillus* was dominant in various autotrophic reactors [40,41]. Furthermore, some species of the genus *Hydrogenophilaceae_unclassified* (abundance 4.53%) use various inorganic electron donors such as reduced sulfuric compounds or hydrogen [42]. As autotrophic denitrifying bacteria, the relative abundance of *Thiobacillus* (from 23.9% to 17.3%) and *Hydrogenophilaceae_unclassified* (from 4.53% to 3.7%) was decreased markedly with the addition of organic matter at the end of stage I. These results indicated that a low concentration of organic matter could reduce the enrichment of autotrophic denitrifying bacteria, which is consistent with a previous study that demonstrated that the operational performance of the autotrophic system was greatly affected by organic carbon [43]. Similarly, the decrease in autotrophic denitrifying bacteria at the end of stage I was responsible for the deterioration of the NRE. However, it should be noted that the relative abundance of the genus *Denitratisoma*, affiliated with heterotrophic denitrifying bacteria, was also decreased (from 8.1% to 4.8%) with the addition of organic matter. This may result from the differential carbon sources, since *Denitratisoma* has been shown to utilize internal carbon sources (i.e., dying cells) [44]. Moreover, *Denitratisoma* has been frequently detected in autotrophic reactors such as A/SAD systems [45], autotrophic denitrification systems [46], and anammox systems [47]. As the single anammox bacteria included in this study, *Candidatus Jettenia* decreased as expected (from 7.1% to 4.5%) with the addition of organic matter, indicating the inhibitory effect of organic matter on anammox bacteria.

The analysis of the relative abundance of denitrifying bacteria revealed that denitrification was strengthened in stage II. In this stage, the relative abundance of *Thiobacillus* and *Denitratisoma* was increased (reaching 28.1% and 8.3%, respectively), indicating that the autotrophic and heterotrophic denitrification processes were simultaneously enhanced with the improvement of the S/N ratio. It is noteworthy that previous studies demonstrated that excess nitrite can inhibit anammox in a SAD system [25,27]. Similarly, increases in the CEA and S/N ratio accelerated the enrichment of denitrifying bacteria in our reactor. Furthermore, accumulated nitrite was utilized by *Thiobacillus* and *Denitratisoma* rather than anammox bacteria. Thus, the relative abundance of *Candidatus Jettenia* was slightly decreased to 3.6% in stage II.

## 4. Conclusions

The performance of A/SAD/HD system was closely related to the CEA and COD in the influent; the analysis of the NRE showed that, under optimal influent conditions (CEA 55%, S/N 1.24, COD 50 mg/L), the NRE reached 95.0% ± 0.5%, this result demonstrated that the A/SAD/HD system can effectively remove total nitrogen. The S/N ratio regulation strategy was feasible when the influent COD concentration was less than 100 mg/L and the CEA was between 57% and 63%. The improved nitrogen removal performance of the A/SAD/HD system compared to sole anammox system was ascribed to cooperation between anammox and denitrification, which was further confirmed by batch experiments and microbiological analysis. 

This study analyzes the effects of CEA and COD on A/SAD/HD systems. However, in actual engineering applications, the influence of environmental factors (temperature, pH, and HRT) on the performance of the system must be considered, and more research is needed to be done.

## Figures and Tables

**Figure 1 ijerph-17-02200-f001:**
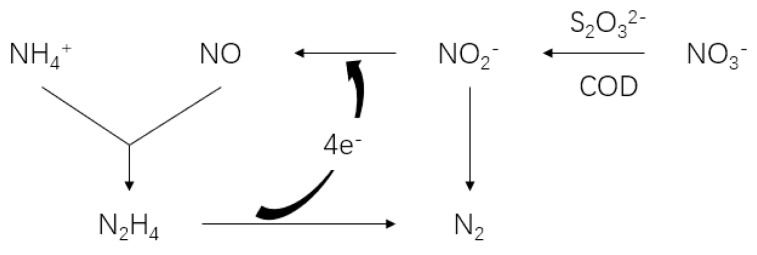
Operational diagram of the anammox, sulfur-based autotrophic denitrification, and heterotrophic denitrification (A/SAD/HD) coupling system.

**Figure 2 ijerph-17-02200-f002:**
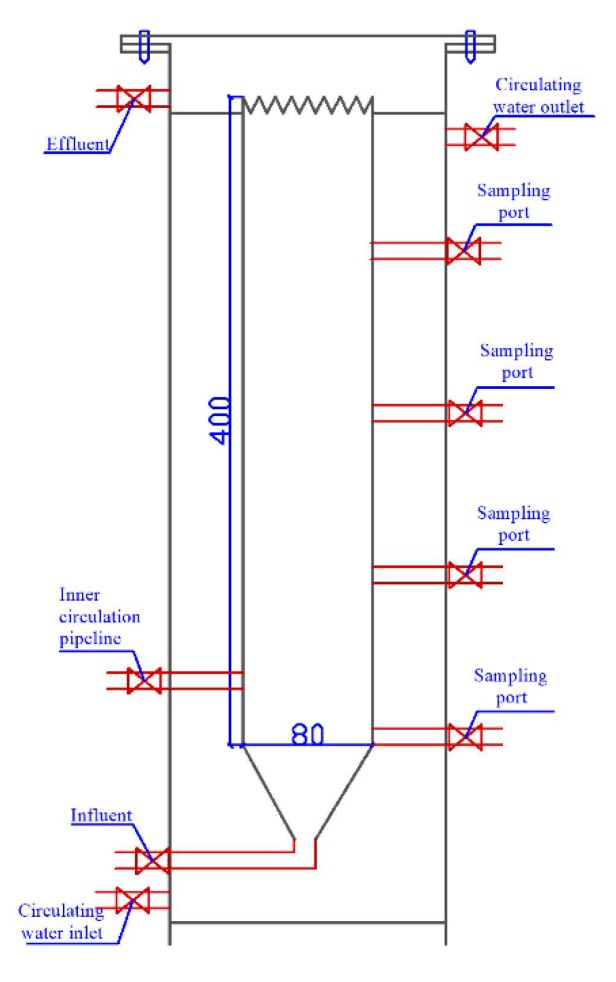
Graphic scheme of the reactor.

**Figure 3 ijerph-17-02200-f003:**
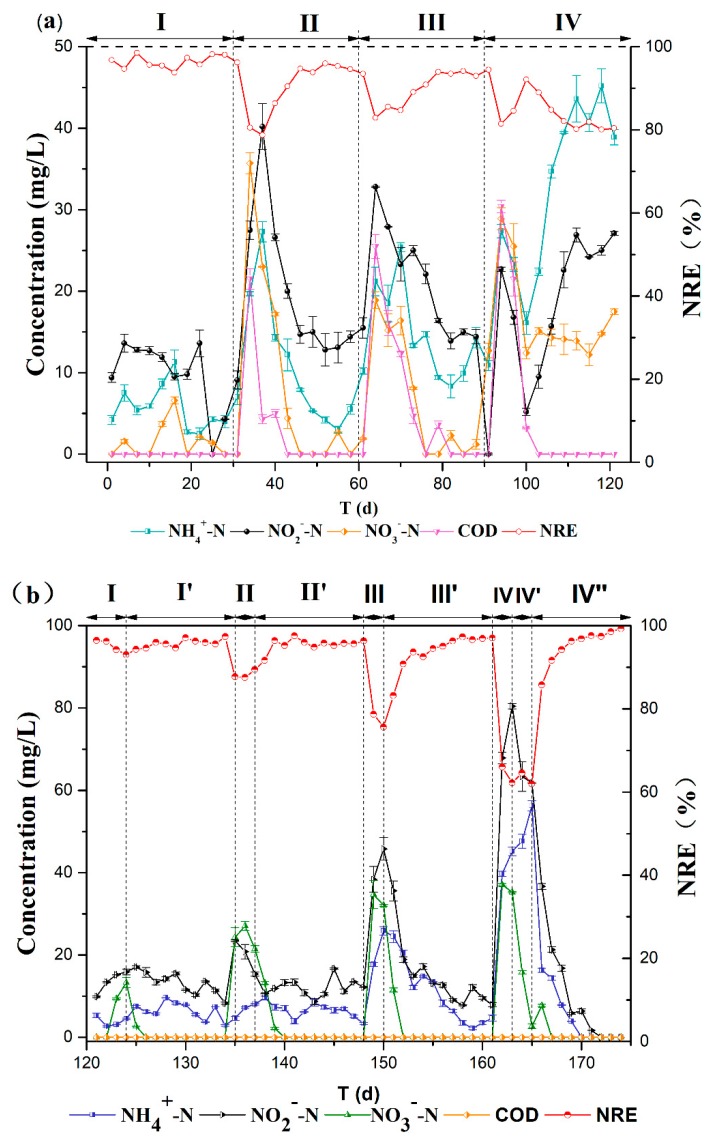
Operating performance of the A/SAD/HD system at stage I (**a**) and stage II (**b**); COD: chemical oxygen demand; NRE: nitrogen removal efficiency.

**Figure 4 ijerph-17-02200-f004:**
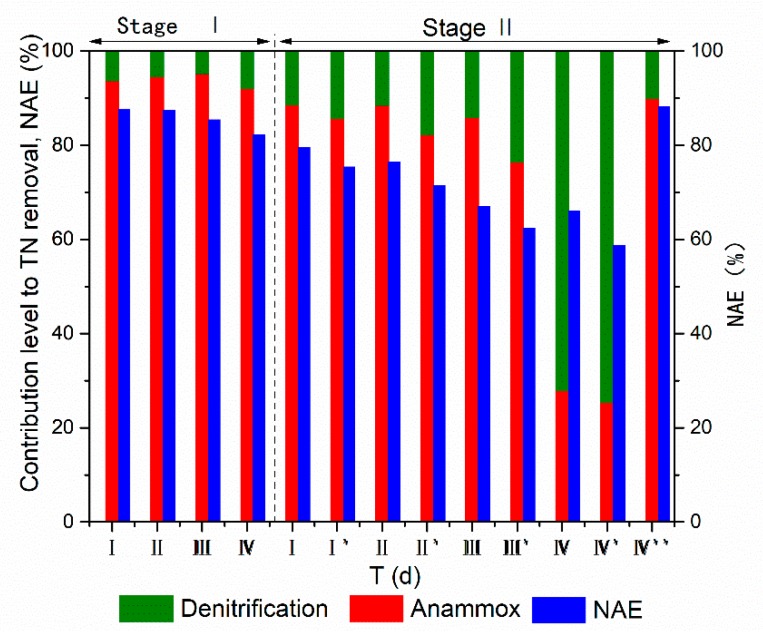
Nitrite accumulation efficiency (NAE) and contribution rate of anammox and denitrification to total nitrogen (TN) removal.

**Figure 5 ijerph-17-02200-f005:**
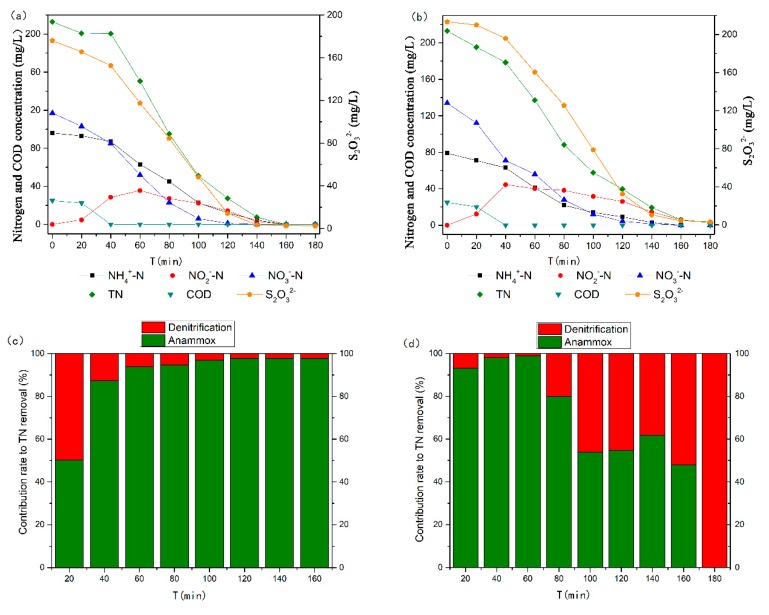
Dynamics of nitrogen removal and thiosulfate consumption and the contribution rates of denitrification and anammox to TN removal in batch experiments: (**a**,**c**) CEA 55%, S/N 1.24, COD 25 mg/L; (**b**,**d**) CEA 63%, S/N 1.38, COD 25 mg/L.

**Figure 6 ijerph-17-02200-f006:**
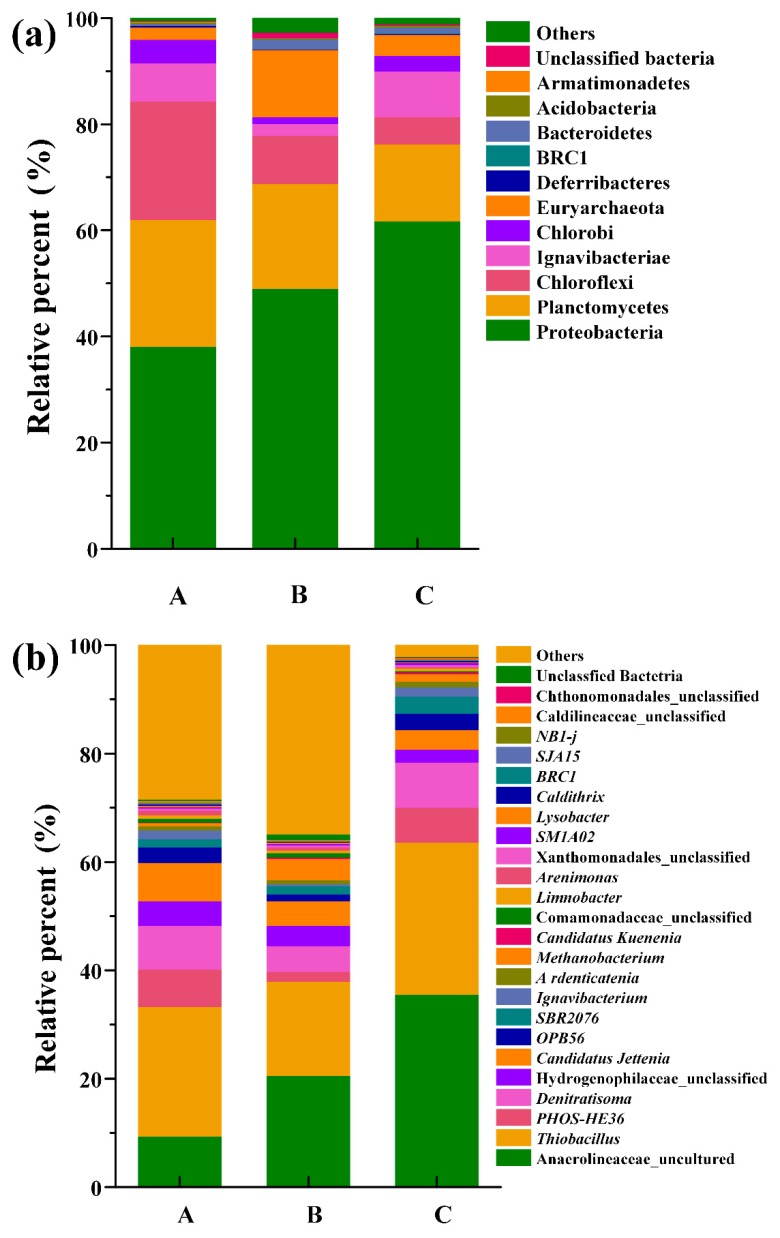
Variations in microbial community structure in the A/SAD/HD reactor at the phylum (**a**) and genus (**b**) levels. (A: the end of phase I (stage I); B: the end of stage I; C: the end of stage II.)

**Table 1 ijerph-17-02200-t001:** Operating conditions of the expanded granular sludge bed (EGSB) reactor in stage I and stage II.

Phases	Stage I				Stage II								
	I	II	III	IV	I	I′	II	II′	III	III′	IV	IV′	IV″
Days	1–30	31–60	61–90	91–120	121–124	125–135	136–137	138–148	149–150	151–161	162–163	164–165	166–174
NH_4_^+^-N (mg/L)	192	192	192	192	175	175	166	166	158	158	149	149	64
NO_3_^−^-N (mg/L)	234	234	234	234	251	251	260	260	268	268	277	277	78
CEA (%)	55	55	55	55	59	59	61	61	63	63	65	65	55
COD (mg/L)	0.00	50.00	100.00	150.00	50.00	50.00	50.00	50.00	50.00	50.00	50.00	50.00	50
S/N	1.4	1.24	1.08	0.92	1.24	1.3	1.24	1.34	1.24	1.38	1.24	1.42	1.24

CEA = NO_3_^−^-N_inf_/(NO_3_^−^-N_inf_ + NH_4_^+^-N_inf_) × 100; COD = Chemical Oxygen Demand; S = S_2_O_3_^2−^; N= NO_3_^−^-N.

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
