# Peer review of "Feasibility of Adjusting the S2O32−/NO3 Ratio to Adapt to Dynamic Influents in Coupled Anammox and Denitrification Systems"

_ijerph, 2020, doi:10.3390/ijerph17072200_

Round 1

Reviewer 1 Report

This paper discusses how the adjustment of sulphur to nitrogen ration can be a feasible approach for adaptation of wastewater influents in a coupled annamox - denitrification systems. This is a topic is of interest to a wide audience of scientists and engineers.

I can suggest publication of the present manuscript if the following minor revisions are addressed:

Title: Acronyms, symbols etc. are not generally used in titles

Line 27: please provide references

Line 44: what is meant by "acceptable"? according to which legislation standards?

Lines 60:66: I suggest the authors should consider a rewording of this paragraph and clearly state the research questions

Line 70: please provide a graphic scheme of the reactor

Line 79: why were these time points chosen?

Line 81: I understand the convenience of calculations, but are these values representative of field conditions? any reference?

Lines 94:99: this paragraph actually belongs to Results section

Line 106: please provide the actual value of concentration

Line 112: please specify the brand and other technical details

Line 113: 'spectrophotometric method' is a very generic term. Please provide more details.

Line 114: same as above

Line 116: please specify the brand

Line 118 (equation 5): it is a bit difficult to read these equations. Please consider report the concentrations in squared brackets and then use "inf" and "eff" as subscripts

Eq. 6: this is a repetition of the previous line. Can authors please consider a reformulation with acronyms?

Lines 125:126: reference?

Lines 146 (and later in the entire text): please report standard errors

Line 151: please provide references

Line 158: are these speculations or have the authors actually investigated what is reported in the following paragraph?

Line 160: this is the first time this species is mentioned. Is C. jettenia an heterotrophic denitryfying bacterium? It is not clear from the way this paragraph is written.

Fig. 2: where are all the standard errors?

Fig. 5: the words are too blur to be read

Line 296: please add 'spp.' or the full name

Line 298: was dominated or was dominant?

Line 313: please consider reformulate this part with terms like ' as expected' or similarly. It might be obvious for the authors but not for all the readers.

Lines 325: 333: This paragraph it is not a conclusion section. It only provides a summary of the main results and does not give any insights from the authors. Why were the results important? what was novel? how can these results advance our knowledge in the field? what should future research should focus on? how can these results be of use in field applications?

Finally, I suggest a final round of text edition with particular focus on punctuation. This will facilitate all the passages to be read clearly.

Author Response

Dear Reviewer:
Thank you for your comments concerning our manuscript . Those comments are all valuable and very helpful for revising and improving our paper, as well as the important guiding significance to our researches. We have studied comments carefully and have made correction which we hope meet with approval. Please see the attachment.

Reviewer 2 Report

The work outlines the study of combine process of Anaerobic Ammonium Oxidation (Anammox) and denitrification systems for improving denitrification by adjusting S/N ratio. In this study, authors used wastewater sludge from leather wastewater treatment plant in Xiangcheng, Henan, China. Authors reported that the component concentration of sludge macroelements and microelements and also reported leading anammox bacteria, Candidatus Brocadia is growing with time and reaches 3.44% to 10.0% within 208 days (supplementary file). But leading bacteria mentioned by authors in the manuscript are Candidatus Jettenia. I hope this will be corrected. It is well known that wastewater sludge with toxic heavy metal ions might be having most inhibition effects on Anammox. Authors should include those sentences in the introduction section about the inhibition effects of metal ions, and if possible, include a control experimental results using a sludge bed with only Candidatus Jettenia bacteria. In the result and discussion section, I found less clarification and careless representation in the supplementary file. Paper has some novel information.

Therefore, I can recommend the present manuscript for publication after a minor revision.

My major issues are as follows:

  1. Line 26; Please write some wastewater treatment technologies for complete nitrogen treatment.
  2. Line 27; Why is anammox among the most promising technologies? Write down the essential points, please.
  3. Line 50; How does sodium acetate exhibit greater efficacy? Include it Please.
  4. Line 57; What is NRE? Please write its full name.
  5. Line 75; need deleting bracket (VSS)).
  6. Please include methods S1 and S2 in the materials and methods section.
  7. Line 92; What are the various condition of CEA? please include it in the manuscript.
  8. Line 99; please include the results of NO2- -N in Table 1.
  9. Line 113; please include the name of the spectrometric method.
  10. Line 171; “…autotrophic denitrifying bacteria, inducing the decreased consumption of sulfide electronic donors” What is the consumption of sulfide electron donors?
  11. Line 221; in ref. 31, authors reported that “COD concentration over 300 mg l(-1) was found to inactivate or eradicate ANAMMOX communities” but in this manuscript reported COD value is less than 100 mg/L is feasible for S/N ratio strategy. Therefore, please include an explanation regarding the best performance of A/SAD/HD system.
  12. Line 239; bracket colour should be removed from [32].
  13. Resolution of Figure 3 and Figure 4 should be increased.

Author Response

(The authors gave the same response as above.)
